# The Power of Stress: The Telo-Hormesis Hypothesis

**DOI:** 10.3390/cells10051156

**Published:** 2021-05-11

**Authors:** Maria Sol Jacome Burbano, Eric Gilson

**Affiliations:** 1Institut for Research on Cancer and Aging, Université Côte d’Azur, CNRS, Inserm, Nice (IRCAN), 06107 Nice, France; maria-sol.jacome-burbano@univ-cotedazur.fr; 2Department of Medical Genetics, Archet 2 Hospital, FHU Oncoage, CHU of Nice, 06107 Nice, France

**Keywords:** telomeres, hormesis, stress response, adaptation

## Abstract

Adaptative response to stress is a strategy conserved across evolution to promote survival. In this context, the groundbreaking findings of Miroslav Radman on the adaptative value of changing mutation rates opened new avenues in our understanding of stress response. Inspired by this work, we explore here the putative beneficial effects of changing the ends of eukaryotic chromosomes, the telomeres, in response to stress. We first summarize basic principles in telomere biology and then describe how various types of stress can alter telomere structure and functions. Finally, we discuss the hypothesis of stress-induced telomere signaling with hormetic effects.

## 1. Introduction

To survive and reproduce, living organisms must maintain homeostasis both in unchallenged (normal) and challenged (stressful) contexts. This requires the evolution of powerful stress response mechanisms adapted to a particular ecosystem and to regular environmental fluctuations. Thus, these mechanisms may be very diverse within the tree of life. The pioneering work of Miroslav Radman on the stress response in bacteria demonstrated the rapid and adaptive value of changing mutation rates for rapid evolution (the mutator effect) [1,2]. In other words, to facilitate the survival of a species, whether it be to respond to a replication blockade or to a stressful environment, it is better to rapidly evolve by generating more mutations, some being possibly lethal, than to die immediately. We believe that this principle applies to the complex dynamics of telomeres in eukaryotes, which become altered in response to stress. Contrary to the general view that these telomere changes are deleterious and inspired by Miroslav Radman’s concepts, we discuss here the possibility that telomere modification in response to stress was selected for during evolution to respond to contextual fluctuations and physiological demands.

## 2. Telomeres: Basic Molecular Mechanisms in an Evolutionary Context

Telomeres are nucleoprotein structures at the ends of linear chromosomes that are required to maintain genome stability by providing a terminal cap [3]. Their function and replication require a complex interplay between specialized and general factors involved in replication and the DNA damage response. Strikingly, their structures change during an organism’s lifetime, both as a result of developmentally regulated programs and external factors. This, together with significant variability among organisms in telomere structure and regulation, raises several fundamental questions regarding telomere evolution and their contribution to development, life history, aging, disease susceptibility and the stress response. To understand fully the “telomere logic” in the stress response, it is necessary to understand the molecular challenges that these structures must face and the evolutionary solutions adopted by different organisms.

### 2.1. Challenge 1: The End-Stability Problem

In eukaryotes, the linearity of chromosomes requires the need to distinguish natural DNA termini from accidental breaks, to avoid unwanted recombination, karyotype instability and unwanted cell cycle arrest. The solution to this stability problem is the specific association of chromosome termini with protective factors including non-coding RNA (Telomeric Repeat-containing RNA or TERRA), and specialized proteins, for which the shelterin complex represents the paradigm [3].

In most eukaryotes, the telomeric DNA sequence is composed of G-rich DNA repeats ending at chromosome termini with a single-stranded DNA G-rich 3′-overhang (or G-tail) [4,5,6]. This 3′ overhang allows the formation of a telomeric DNA loop (t-loop) resulting from the invasion of the 3′ overhang into the internal duplex DNA. The formation of this t-loop is required to maintain telomere stability in a large range of organisms and in many different types of cells. Its formation is assisted by the shelterin subunit TRF2 [7].

The shelterin complex consists of proteins bound to the double-stranded part of telomeric DNA (e.g., TRF1 and TRF2 in chordates; Rap1 in *Saccharomyces cerevisiae*; DNT-1 and DNT-2 in *Caenorhabditis elegans*) [3,8] and single-stranded telomeric DNA (e.g., POT1 in humans; POT-1/POT-2 in nematodes). Finally, some shelterin subunits create a bridge between the duplex and the 3′ overhang nucleoprotein shelterin complexes (e.g., TPP1/ACD and TIN2 in mammals), or simply bind to specific other subunits (for example, mammalian RAP1 binds to TRF2) (Figure 1). One notable exception to this scheme is *Drosophila melanogaster* telomeric DNA, which is composed of long arrays of the non-long-terminal-repeat retrotransposons HeT-A, TART and TAHRE, bound by the Terminin protein complex that has no clear evolutionary relationship with shelterin and is elongated by retrotransposition, but also by recombination and gene conversion [9,10].

### 2.2. Challenge 2: End Replication Problem

The full replication of the terminal DNA cannot be achieved by conventional replication machinery [19,20,21,22]. On the lagging telomeres, the G-tail formation may result simply from the removal of the last RNA primer. On the leading telomeres, the end product is either blunt or 5′ protruding. Since in many organisms a G-tail can be detected at both daughter telomeres, a 5′ resection activity must take place to convert it into a 3′ overhang. Indeed, coupled with replication, TRF2 targets the leading telomere with the 5′-exonuclease Apollo, which initiates the 5′ resection. At the lagging strand, POT1 inhibits Apollo resection [23]. At both ends, the Exo1 nuclease elongates the 3′ overhangs by pursuing a 5′ resection. It remains unknown how this resection is controlled. Finally, POT1 recruits the single-stranded DNA-binding trimeric complex CST (CTC1-STN1-TEN1) to refill the 5′-strands [23] (Figure 1).

There are several processes that compensate for this unavoidable shortening. One such mechanism relies on telomerase, a specialized reverse transcriptase with an RNA template that elongates the G-rich strand using the 3′ overhang as a substrate. Telomerase is composed of two subunits: TERT, the telomerase reverse transcriptase, and TERC, the telomerase RNA template. Using a single telomere length system and monitoring telomere DNA length (TL) evolution in budding yeast, it was shown that telomerase activity is reduced by over elongated telomeres: excess of Rap1p at the long yeast telomeres triggers a progressive *cis*-inhibition of telomerase [24]. This indicates that it is important to keep TL within a regulated range of size.

In mammalian species with large body mass, including humans and some birds, the level of telomerase expression is much higher in germ, stem and progenitor cells than in somatic differentiated cells. In some species, such as laboratory mice, telomerase expression is less repressed and more evenly distributed within cell types than in humans, but still higher in the germ and stem cell compartments [25,26,27]. Importantly, the amount of telomerase present in adult somatic stem cells and progenitors is generally not enough to replenish telomeric DNA fully when these cells divide to regenerate the corresponding tissue or to respond to external stimuli.

In many vertebrates, TL is subjected to a progressive erosion cadenced by cell divisions and the residual level of telomerase activity. There are also exceptions to this scheme; for instance, in the long-lived bat species *Myotis myotis*, mean TL does not shorten with age but changes with the environment [28]. In fact, the rate of telomere shortening in blood cells during development and aging is not strictly related to the level of telomerase activity [6,29]. This suggests that telomerase may have functions other than regulating telomere length and that TL might be regulated independently of telomerase. Indeed, the mean TL is the net result of shortening and elongation rates, rendering it difficult to interpret TL changes only in terms of telomerase-dependent elongation [24]. One interesting hypothesis is that the rate of telomere DNA shortening, but not the absolute length, controls the aging process [29]. It is not yet known whether there exist developmentally regulated mechanisms, independently of telomerase activity, that modulate the rate of TL shortening.

Interestingly, eukaryotic cells are able to use a telomerase-independent mechanism, ALT (alternative lengthening of telomeres), which is based on homologous recombination [30]. It is activated in a number of human tumors, in human cells immortalized in culture, and also in normal somatic tissues [31]. In plants, the ALT mechanism is activated upon telomerase dysfunction and possibly also during the earliest stages of normal plant development [32]. At the organismal level, ALT was suggested by Gomes and colleagues [6] for koala and wombat telomeres, based on the presence of long and interrupted telomere repeats and the absence of telomerase expression. However, it is not known whether these ALT-like telomeres use the same process of maintenance as the ALT + cancer cells. Along this line, a recent work that revealed the existence of an ALT-like mechanism in *Terc* knock-out (KO) mouse stem cells, with a unique non-telomeric sequence (mTALT) [33]. This process might be related to the one used in wombat. Hence, ALT or ALT-like mechanisms could give an advantage in the context of species with no telomerase. In order to avoid cancer development, we can imagine ALT process is finely regulated in species using only this pathway to preserve their telomeres.

### 2.3. Challenge 3: The Hard-to-Replicate Problem

In addition to the end-replication problem, the progression of the replication fork through telomeric chromatin is difficult and may result in fork collapses leading to the loss of a large part of the telomeric DNA. This is prevented by specific mechanisms involving the shelterin subunits TRF1 and TRF2: TRF1 manages classical stalled forks leading to ATR activation and TRF2 manages topologically-constrained stalled forks [18,34] (Figure 1).

Interestingly, the function of TRF2 in this process requires the same factor, Apollo, as the one favoring the progressive shortening of the leading strand [18]. Inhibiting TRF2-Apollo interaction is associated with severe progeroid syndrome in humans [35]. The long-lived giant tortoises *Chelonoidis abingdonii* and *Aldabrachelys gigantea* carry an Apollo variation within the TRF2 binding domain that is not conserved in other reptiles, fishes or mammals and, thus, could contribute to the extreme longevity of these organisms [36]. Altogether, these findings suggest an evolutionary advantage, at least in some organisms, to control progressive shortening due to the end replication problem. This hypothesis seems at odds with the wealth of studies showing that TL is positively associated with improved healthspan and lifespan in vertebrates [37,38]. Usually, short telomeres are associated with chronological aging, cardiovascular and age-related diseases and poor prognosis [39]. However, long telomeres are not as beneficial as was initially thought: several studies have shown an association between long telomeres and some types of cancer. Actually, in a genome-wide association study, nine cancers out of 35 (including glioma, neuroblastoma and endometrial cancer) were associated with increased TL [40]. To understand the “raison d’être” of these different telomere trajectories between organisms and individuals, and their consequences for health, we need to understand more fully the complex interplay between genetic and environmental determinants regulating the response of telomeres to stress during development and aging.

## 3. Telomere Response to Stress

A stressor can be defined as a factor that leads to a rupture in homeostasis, whether it be at the cellular, systemic or organismal level. Thus, the environment can be defined as the combination of stressful situations to which individuals are exposed during their lifetime, including lifestyle, climate, pollutants, pathogens and social relationships. Organisms have evolved mechanisms to cope with their environments and predictable environmental fluctuations. Abnormal exposure to stressors is a well-known risk of age-related disease and premature aging [41]. Different types of stressors also lead to an acceleration of TL shortening, leading to critically short telomeres and cellular senescence, a mechanism that could explain the link between stress and aging. To give an example, the telomere shortening rate can increase by roughly five times in fibroblast cultured in hyperoxia compared to normoxia [42]. The rapidity by which these changes are signaled to the cells is not well understood. One can imagine two non-mutually exclusive modes of signaling. On one hand, shortened telomeres are phenotypically “silent” until they reach a critically short length that triggers senescence. On the other hand, even small changes in telomere length variation can have an impact, even before cells reach senescence. In the following paragraphs, we discuss how various stresses can modulate telomere structure and function (Figure 2).

### 3.1. Inflammation

During aging there is an increase of circulating inflammatory markers and a decrease in immunological capacities, leading to a systemic chronic inflammatory state, also known as inflammaging [44]. Chronic inflammation is often associated with diseases such as rheumatoid arthritis or chronic obstructive pulmonary disease (COPD). Noteworthy, liver TL attrition is significantly exacerbated in cirrhosis or chronic hepatic livers [45]. In mouse, repeated *Salmonella* exposure decreases TL in leukocytes, but not in hepatocytes or splenocytes [46]. If these results indicate that inflammation can drive TL shortening, the reverse can also occur and a wealth of data present telomere dysfunctions as drivers of chronic inflammation (see e.g., [47].) In this review, we will focus on the processes of telomere shortening triggered by inflammation.

Inflammation-induced TL shortening may be due to the inhibition of telomerase (Figure 2). In support of this hypothesis, TNFα treatment shortens telomeres via ATF7-dependent telomerase disruption [48]. However, infection is capable to trigger accelerated TL attrition in telomerase-deficient mice [49], revealing the existence of telomerase-independent mechanisms linking inflammation to TL shortening. These mechanisms might include the production of ROS during inflammation; for example, via secretion by immune cells such as neutrophils [50]. Indeed, the rate of telomere shortening increases under oxidative stress, and the G-richness of telomeric DNA renders it highly susceptible to oxidization, leading to the accumulation of 8-oxo-guanine. This disrupts the binding of telomere protective factors and can induce DNA breaks and prevent extension by telomerase (Figure 2) [51,52].

### 3.2. Life Factors

Under psychological stress, the Hypothalamic-Pituitary-Adrenal (HPA) axis secretes high levels of glucocorticoids (GC): cortisol and corticosterone. Their release allows an immediate response to stress, but chronic exposure may be detrimental [53]. GC block certain non-essential processes to store energy during periods of stress [54] but are also associated with behavioral response. There are several examples of telomere changes due to psychological stress. For instance, TL shortens in children who experience a violent childhood [55]. Acute psychological stress also increases telomerase activity for short periods, correlated with cortisol/corticosterone levels in humans and rats [56,57]. In fact, glucocorticoid accumulation seems to modify telomere homeostasis, including TL, although the mechanisms involved remain elusive [58]. One possible mechanism links psychological stress and oxidative stress, which is one of the consequences of GC activity [59]. Indeed, GC regulating genes and downstream pathways are involved in ROS production (Figure 2) [54] and/or mitochondrial metabolism [60]. Highly stressed women have shorter TL, decreased telomerase activity and increased oxidative stress signatures [61]. Thus, oxidative stress may be a mechanism for the effects of psychological stress and GC on telomeres, as it is for inflammation.

Other daily way of life decisions, such as diet, smoking and physical activity have been linked to telomeric changes. To give an example, the Mediterranean diet has been highlighted by its anti-aging capacities and cohort studies suggest that it deaccelerates telomere shortening in humans [62]. Indeed, individuals suggested to a diet rich in fibers, vitamins and unsaturated fatty acids tend to have longer telomeres and lower oxidative stress. Caffein consumption, another strong antioxidant agent, has also been correlated with longer leukocytes TL [63]. Food can modulate TL but also telomeric protein state. For example, Chinese obese children had shorter TL but higher TERT promoter methylation [64]. Additional to food intake, several studies have analyzed an association between exercise and telomere length. To our knowledge, most of the published results show a positive correlation between TL and training, as compared to sedentary people [65]. Nevertheless, physical exercise might also lead to a certain level of telomere dysfunction: occasional intense exercise would tend to produce an excessive amount of ROS leading to higher DNA damage, and thus, telomere damage while low intensity but regular activity would lead to lower oxidative stress and better adaptation [66]. Physical exercise can also modulate telomere structure and function by increasing TERRA transcription [67].

### 3.3. Chemical Stress

Exposure to chemical stressors can greatly modulate the health of organisms. The stressors can be toxic/genotoxic compounds or pesticides and airborne pollutants. Several studies report an association between chemical exposure (benzene, polycyclic aromatic hydrocarbons, ethanol, etc.) and telomere attrition in human cells [68,69]. Ethanol triggers TL shortening in human fibroblasts, whereas in budding yeast it has a TL lengthening effect [69,70]. In humans, the shortening effect of ethanol appears to be mediated by oxidative stress (Figure 2), mainly induced by alcohol dehydrogenase [69,71]. Nevertheless, zebra finches exposed to trace metal elements exhibit decreased TL with no change in oxidative stress [72]. Overall, the effect of chemical stress on TL varies depending upon the nature of the chemical, the mode and duration of exposure and the organism exposed. The mechanisms appear to be quite diverse, but in at least some cases are mediated by oxidative stress.

### 3.4. Physical Stress

Physical stress is another source of external tension for telomeres. It is well known that genotoxic insult triggers persistent DNA damage at telomeres [73,74]. A particular case is UV irradiation, whose main photoproduct is cyclobutane pyrimidine dimers formed between two consecutive pyrimidine bases, which are often present in telomeric repeat DNA sequences. Thus, telomeres are a preferential target for UV-induced DNA damage [73]. If the telomeric photoproducts at telomeres remain unrepaired they can interfere with shelterin DNA binding, telomeric DNA replication and transcription, and thereby lead to TL shortening and uncapping [75]. Indeed, UV exposure is associated with accelerated TL shortening [76,77,78] and prevents TRF1 binding [79]. However, another study did not observe increased TL shortening in human fibroblasts irradiated with UVB rays [73]. It is possible that the ability of telomeres to repair UV-induced photoproducts varies among cell types and organisms, which could be interpreted as an adaptive strategy of telomeres to repair UV lesions only when needed.

Recently it was shown in astronaut leukocytes that long-duration spaceflights (6–12 months) lengthened telomeres, but that telomeres were rapidly shortened to a pre-spaceflight length within a few days after the astronauts returned to Earth [80,81]. The authors posited that these effects were caused by space radiation. Puzzlingly, this telomere elongation might not be due to the canonical telomerase activity, which decreases during spaceflight, but to an increased rate of DNA damage and recombination that could favor ALT mechanisms. Indeed, astronaut cells exhibit elevated DNA damage and increased mitochondrial stress and ROS production. One interesting hypothesis is that oxidative stress activates the ALT pathway by triggering replication stress [82] and relocating telomerase to mitochondria (Figure 2) [83].

Exposure to elevated temperatures is another environmental stress that could affect telomere dynamics. Indeed, in *Saccharomyces cerevisiae*, incubation at high temperatures shortens telomeres [70]. Human cells that undergo heat shock accumulate TERRA and disrupt TRF2 from telomeres [84,85].

Finally, pH variation may be a cause of stress within an ecosystem. TL was observed to be reduced in HeLa cells cultured at low pH, but the heterogeneity of individual TL was narrow [86]. The authors proposed that telomerase elongates short telomeres under low pH, while long telomeres are more strenuously eroded.

## 4. Telomere and Life-History Trade-Off

Individual lifespan is influenced by life-history traits such as time of breeding, lifespan and the size of offspring. These life decisions are often highly energy-consuming and are known as life history tradeoffs. Among these trade-offs, the disposable soma theory states that the soma is just a vehicle to preserve the “immortal” germline. An important current debate on telomere biology is to determine whether developmentally programmed telomere change can contribute to life-history. For instance, in several species (e.g., human, rhesus monkey, zebra finch, dog) telomerase is inhibited in somatic cells after embryonic development but its expression is preserved in the germline [6,86,87], suggesting a trade-off between the cost of telomere maintenance in the soma and the need to allocate resource toward reproduction.

A high level of somatic telomerase downregulation appears to have been selected for in long-lived, large-body-mass animals, including mammals [6,88,89]. Accordingly, long-lived mammals have shorter telomeres (14 kb in African elephants or 5–15 kb in humans) than short-lived mammals (over 40 kb in the laboratory mouse). These observations can be interpreted with caution due to the extraordinary diversity in life-history trajectories in the tree of life. Thus, it is important to tackle this question by comparing related species but with different environment and life-history traits. For instance, mouse strains recently derived from the wild have shorter TL than those already established in laboratories. Overall, among these various mouse strains, no correlation exists between TL and lifespan [87]. This was also observed in rats [88]. Thus, to harbor shorter TL appears more related to the wild conditions than to regulate lifespan. In agreement with this view, a recent analysis showed that wild mammals have shorter TL than those that domesticated one [89]. Overall, these studies suggest that possessing short telomeres is advantageous in a wild habitat with limited resources.

Another interesting link between telomere and life history is the correlation between the rate of TL somatic shortening and different strategies of growth and reproduction. Several organisms, including captive and wild metazoans, exhibit a negative correlation between reproduction and TL [90,91,92,93,94,95,96]. For instance, there is evidence that longer TLs triggered a reduced fertility but no effect on lifespan in *Drosophila melanogaster* [97]. In birds, earlier breeding is associated with more successful reproduction but shorter TL. One study found that in the common tern *Sterna hirundo*, brood size is inversely correlated with parental blood TL [92]. The authors reported that each fledgling in the brood shortened parental telomeres to the same extent as four years of normal aging. Telomere shortening appears to be one of the costs of reproduction and a negative correlation between reproduction and longevity has been reported in numerous species (e.g., humans, *Caenorhabditis elegans*) [98,99]. Overall, these findings suggest that TL shortening contributes to accelerated aging as a consequence of reproductive activity. However, there are exceptions to this telomere-mediated reproduction trade-off. For instance, in the eusocial insect *Bombis terrestris*, bumblebee queens live 4 to 6 times longer than workers and exhibit increased telomerase activity in fat bodies [100].

In many species, reproductive success does not rely solely on producing offspring but also on supporting their survival through early life care and breeding. In the Seychelles warbler (*Acrocephalus sechellensis*), a cooperatively breeding species, elderly dominant females that have subordinate helpers exhibit reduced telomere attrition and senescence markers, as well as improved probability of survival [101]. This may be because the dominant females expended less breeding effort due to helper incubation attendance and support for chick provisioning. In the common tern, males take care of feeding chicks and their telomere attrition associated with breeding is greater than that of females [92]. Finally, TL shortening is greater in human mothers with chronically ill children [61], showing that care-related stress is directly associated with telomere dynamics. These results suggest that not only reproduction must be considered in telomere attrition cost but also caregiving for offspring.

The mechanisms explaining these various relations between telomere variations and life-history traits remain largely unknown. For instance, how the energy saved by somatic telomere shortening could be reallocated to reproduction? One challenge in experimentally testing this role of telomeres in the germen/soma tradeoff is the difficulty of quantitatively comparing the energy cost to maintain telomere length and function with breeding function [102]. Telomere initial length and telomere attrition rate vary among species. For example, humans have short telomeres (5 to 15 kb) and their mean lifespan is estimated to be 72.6 years. Therefore, telomeres represent only 0.00016 to 0.0048% of the human genome. If we take into account the telomere shortening rates published in other reports, the base pair loss per year in leukocytes represents only 0.00000226% of human DNA. Laboratory mice have short lives (2 years) but long telomeres (50 kb); they lose just 0.000123% of their genomes every year despite their relatively high telomere attrition rate (7000 bp) [29]. Despite a relatively high level of somatic telomerase activity in mice, they have a rapid telomere erosion rate. Consequently, it is difficult to imagine that the energy saved by not preserving somatic TL could be enough to significantly offset the costs of growth and breeding. In support of this hypothesis, male Cory’s shearwaters with larger offspring displayed shorter TL, whereas highly reproductive females had long telomeres [103]. Importantly, eggs were removed from nests to eliminate parental nesting/breeding effort, but TL values were compared with non-manipulated birds.

In summary, our knowledge of the role of telomeres in different life-history trajectories remains limited and puzzling. In particular, it remains difficult to ascertain whether somatic TL shortening is an evolutionary strategy to allocate resources to reproduction or merely a consequence of breeding and care. It is also possible that the TL shortening cost linked to reproduction serves other functions in life history. If this is the case, it raises the question of whether TL shortening is a cost or a benefit.

## 5. The Telomere Hormesis Hypothesis

If the germ/soma trade-off cannot really explain the evolutionarily conserved process of telomere shortening during the ontogeny and in response to stress, what is its “raison d’être”? Is there any advantage to shorten telomeres upon exposure to stress or upon reproduction? Here, we address this question by using the concept of hormesis that can be defined as a dose response process in which low and high doses of stressors have opposite effects [104]. Applied to telomeres, a hormetic effect could be beneficial in, a priori, two types of situations: i) a low level of dysfunction (shortening or chromatin changes) leading to a beneficial effect, for instance by triggering a redox signaling or changing gene expression, while a higher dose (critical shortening or profound chromatin destruction) would trigger senescence, chromosomal instability and aging; ii) a low level of senescence, triggered by telomere dysfunction, being beneficial (for instance to limit the proliferation of pre-cancerous cells) while too many senescent cells would favor aging and even cancer. Pioneer data from Seiichi Yokoo and colleagues, suggested telo-hormesis effect on human keratinocytes exposed to low doses of hydrogen peroxide [105]. H_2_O_2_ effects regarding the number of population doublings, ROS amounts and TL were better than those observed on keratinocytes exposed to antioxidant agents. Later, this concept was discussed in a study on *cdc13-1* dysfunctional budding yeast, which found that pre-exposure to a low dose of telomere stress led to more resilient yeast under acute stress [106]. A recent report showed that TL was shorter in mammals living in the wild than in domesticated species [89], suggesting that possessing short telomeres in a wild habitat confers a selective advantage (Figure 3). Since a wild habitat forces organisms to cope with both resource scarcity and fluctuating environments, we can postulate that short telomeres may improve survival and fitness during exposure to stress.

### 5.1. Telomere Changes to Prevent Tumor Formation

An experimentally documented advantage of telomere shortening is preventing tumor progression by sending pre-cancerous cells with uncontrolled cell division to senescence. Indeed, oncogenic stress can lead to telomere dysfunction and senescence [107,108]. Moreover, short telomeres are frequently observed in long-lived, large-body-mass species, which are at relatively high risk of developing tumors [6,89,92,109]. Moreover, in Genome-Wide Association Studies (GWAS), mutations in genes favoring telomere elongation have been associated with increased risk of lung cancer and melanoma [110]. Accordingly, protection against melanoma development might be a driver of short TL in European populations compared to Sub-Saharan Africans [111]. These findings raise the question of whether UV damage may distinctly affect DNA from different populations. To our knowledge, some studies have compared the SNPs within DNA damage repair genes among different ethnic [112]. However, whether there is an ethnic difference between UV-induced DNA damage repair in European and Sub-Saharan populations, has not been studied yet.

Corroborating these observations, mice with dysfunctional telomerase are protected from chemical-induced skin cancer [109], while mice overexpressing telomerase develop large tumors relatively quickly [113]. Furthermore, crossing late generations of telomerase-deficient mice with strains prone to developing cancer reduces tumor frequency and increases survival [114,115,116,117,118]. For instance, in a lymphoma mouse model, short telomeres triggered a p53-mediated senescence process that limits tumorigenesis [116]. 

One can imagine that, if telomere attrition is an adaptive tumor-suppressor strategy, individuals with long telomeres should be more prone to cancer. However, Muñoz-Lorente et al. found that a mouse model carrying hyperlong telomeres, but with normal telomerase expression, exhibited lower cancer incidence and longer lifespan [119]. Thus, telomere-related changes other than TL shortening can interfere with cancer formation. For instance, crossing telomerase-dysfunctional mice with colon carcinoma-prone mice (*Apc^Min^*) revealed that short telomeres can promote tumor initiation by inducing chromosome instability, but also prevent tumor progression [115]. In addition to promoting chromosome instability and tumor initiation, telomere dysfunction in stromal cells can favor carcinoma development by creating a pro-inflammatory environment [120,121].

Apart from telomere length variation, modulation in the expression of telomere factors can directly favor tumor formation and progression. For instance, the reactivation of telomerase observed in the large majority of human cancers might mediate pro-oncogenic effects through non-canonical functions [122]. Changes in shelterin proteins can also be drivers of tumorigenesis. Indeed, POT1 variants are correlated with several types of cancer [123,124,125,126,127]. In a large number of human cancers, high levels of TRF2 are associated with poor prognosis, favoring neoangiogenesis and immune-escape [128,129,130,131].

These opposite roles of telomeres in oncogenesis can explain the seemingly contradictory results obtained regarding the link between TL and cancer in the general population (Figure 3). Moreover, they invite us to revisit the view that telomere shortening was selected as an anti-cancer response. According to the pleiotropic antagonism theory of evolution, telomere shortening may limit tumorigenesis during youth but promote it later in life. In this case, the anti-cancer hormetic effects of telomere dysfunction would be restricted to the reproductive period of life. In contrast, in older adults, excessive telomere shortening and checkpoint failure might favor tumor formation both by cell-autonomous and cell-non autonomous mechanisms. Thus, before proposing a pro-telomerase therapy for older adults, it must be determined whether telomere hormetic effects can still be a factor later in life.

### 5.2. A Hormetic Effect of the Telomere Position Effect 

Telomeres play a transcriptional regulation role known as the telomere position effect (TPE) [132]. In short, a consequence of telomere shortening due to stress might be to mount an adaptive transcriptional response (Figure 2). This mechanism has been thoroughly studied in budding yeast: Rap1 binds to double-stranded telomeric DNA, where it recruits the silencing factors Sir2, Sir3 and Sir4, initiating the spread of a silencing complex through subtelomeric chromatin [133]. Thus, TPE depends on both TL and telomere proteins. The extent of its spreading is modulated in a chromosome-end-specific manner. Insulator and proto-silencer sequences are present in the subtelomeres and, through a complex combination of chromatin looping, lead to a discontinuous spreading that can silence genes located far away from telomeres at the yeast chromosome scale [134]. A similar TPE is conserved in *Drosophila*, mammals and Trypanosoma [135,136,137,138,139,140,141]. As in yeast, human TPE is regulated by long-range chromatin loops whose formation depends upon TL and shelterin subunits [142,143,144,145]. The existence of long-range chromatin-loops has first been described in yeast [146,147,148,149,150], then in plant [151] and in mammalian cells [146,147,149,152,153]. Although the mechanisms allowing these long-range interactions remain elusive, it has been shown that human telomeres can interact with interstitial telomeric sequences, or ITS, in a Lamin A/C and TRF2 dependent manner [145,154]. This chromatin looping is expected to have a direct impact on the transcriptional regulation of the associated subtelomeric genes [132]. TL shortening can decrease the strength of TPE, thus upregulating subtelomeric genes, while releasing telomere factors that bind to internal sites, where they regulate the expression of another set of genes. This complex transcriptional regulation mediated by TL changes was first described in budding yeast [155,156] but also occurs in mammalian cells [157].

Remarkably, the genes whose expression is regulated by TL (such as those involved in carbon source and nutrient utilization [152], cell wall and stress responses [153,158,159]) can be adaptive in response to stress because they are in a subtelomeric position or targeted by released telomere factors. For instance, subtelomeric derepression occurs in response to glucose starvation via a global decrease in histone levels [160], and the derepression of subtelomeric genes encoding cell wall components contributes to the adaptation of yeasts to the membrane-deforming molecule chlorpromazine [158]. Upon rapamycin treatment, Sir3 is hyperphosphorylated, subtelomeric silencing is reduced and yeast acquired greater stress resistance. Besides, still in budding yeast, mitochondrial ROS are sensed by DNA damage kinases to extend chronological lifespan. This mechanism is mediated by enhanced subtelomeric gene silencing through the removal of the histone demethylase Rph1 favoring the Sir3 spreading in subtelomeric regions [161]. In several human cell lines, the ubiquitin-like protein ISG15 (interferon-stimulated gene 15) is overexpressed upon telomere shortening. *ISG15* is involved in the innate immune anti-viral response [162]. Altogether, these results substantiate the telo-hormesis concept, showing a direct telomere adaptation in response to stress via TPE modulation (Figure 3).

Beyond yeast and mammals, clustering stress response genes at subtelomeres appears to be a widely conserved strategy throughout evolution. For instance, in humans, a family of olfactory receptor genes is located at subtelomeres [163]. However, it remains unknown whether their expression is regulated by telomeric changes. Remarkably, a similar strategy is used by *Candida glabrata* and *Trypanosoma brucei* to regulate the expression of subtelomeric variant surface glycoprotein (VSG) genes, to evade the host immune system [141,164]. The link between telomere changes and the precise regulation of VSG gene expression is unclear, but it may be related to stress signals emanating from the host.

In order to prevent activation of mobile DNA elements, subtelomeres, as other repetitive DNA sequences, are usually transcriptionally repressed. For subtelomeres, this repression is mediated by TPE. Regarding the telo-hormesis hypothesis, the concept of dose-response is well conserved in TPE maintenance. In fact, modest TL shortening triggers derepression of advantageous stress-response genes, while chronic TL changes can relieve TPE and activate mobile DNA elements leading to high-order disorganization and instability.

### 5.3. Telomeres and Mito-Hormesis

The most studied pathway of hormesis is mito-hormesis, defined as mitochondrial stress leading to better health and viability [165]. Mitochondrial dysfunction is signaled by the nucleus through a retrograde communication pathway triggering a differential gene expression response. Among the mito-hormesis signals of stressed mitochondria, the major factors are ROS, but there are also metabolites and the mitochondrial unfolded protein response (UPR).

There is a wealth of data showing a strong link between telomeres and mitochondria (Figure 3). Much data indicate that mitochondrial dysfunction leads to telomere stress and *vice versa*. Telomeric repeats are preferential targets of oxidative stress-induced DNA damage, due to its G-rich sequences [52], a process dubbed teloxidation [43]. Telomere shortening is exacerbated by oxidative stress [166] and enhanced antioxidant contexts tend to lengthen TL [167,168]. Some antioxidant proteins are associated with telomeres (PRDX1 and MTH1) [169,170]. Recently, it was shown that the ROS secreted by activated neutrophils leads to teloxidation of neighboring cells and senescence [50]. Besides, extra telomeric functions of the telomerase subunits TERT and *TERC* have been related to modulating mitochondrial metabolism. First, upon stress, TERT is exported from the nucleus to the mitochondria, where it protects from DNA damage, apoptosis and oxidative stress [171,172,173,174,175]. Second, the telomerase RNA *TERC* is processed within the mitochondria to be later released again to the cytoplasm [176]. It is likely that cleaved *TERC* inform the nucleus about mitochondrial functional state and activity [175]. In telomerase-deficient mice, short telomeres induce p53 activation, which in turn directly represses *PGC-1α* and *PGC-1β*, increasing ROS production. Since ROS signaling is the main mechanism of mito-hormesis and since dysfunctional telomeres could trigger mitochondrial instability and ROS production, telomere-induced oxidative metabolism changes could have a hormetic effect.

Moreover, KO of each shelterin subunit show that they are all, except from TRF1, involved in mitochondrial metabolism [177]. Specially, the glycolysis and the pentose phosphate pathways were modulated upon shelterin KO. It suggests that telomere dysfunction influences metabolism to better adapt the energy consumption upon stress. Interestingly, TIN2 can be localized within the mitochondria and affects it activity and ROS production [178]. TIN2 specific KO affects metabolites of the tricarboxylic acid cycle (TAC), supporting its role in mitochondrial function. Additionally, in human and mouse skeletal muscle, the expression of the mitochondrial sirtuin *SIRT3*, a subtelomeric gene, is activated by TRF2 fixation at its locus through a long-distance chromatin loop. SIRT3 is downregulated upon TRF2 inhibition, leading to mitochondrial dysfunction, characterized by higher mitochondrial DNA content and ROS levels [145]. TRF2 downregulation in skeletal muscle occurs naturally during young adulthood. Is the mitochondrial dysfunction due to TRF2 downregulation an adaptive state to respond to physiological demand and exercise? (Figure 3).

Overall, telomere and mitochondria are functionally interconnected. Telomeric changes are likely to be precursors, at least in part, of the mito-hormesis response. Further studies are required to reveal the mechanisms of this telomere/mitochondria cost/benefit balance and its implications in response to stress.

## 6. Conclusions

Telomere changes may be considered deleterious when they lead to telomere shortening and mitochondrial dysfunction but, in certain circumstances, they could also be part of hormetic tradeoff, improving stress resilience and adaptation to adverse situations. This process appears to be conserved across evolution. Clearly, further studies are needed to explore the mechanisms, the inducers and the evolutionary advantages of short TL to substantiate the telo-hormesis hypothesis. One interesting possibility may be that programmed telomere changes play a role during development via a hormetic effect. 

## Figures and Tables

**Figure 1 cells-10-01156-f001:**
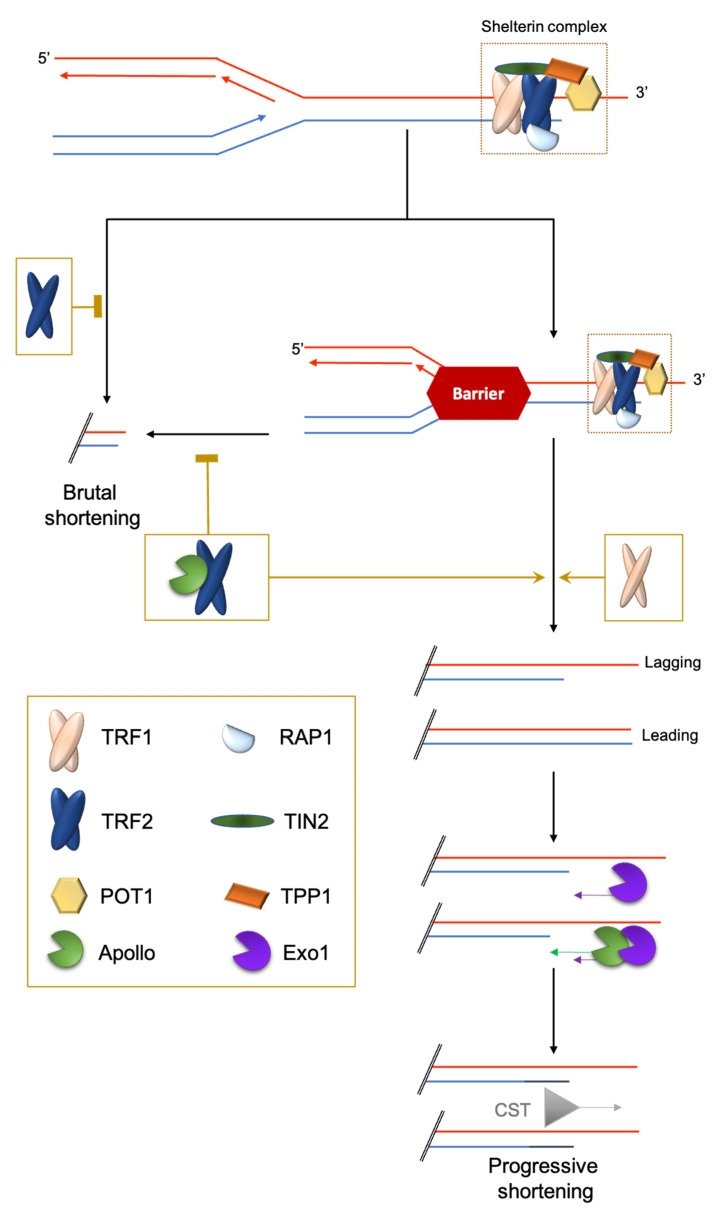
Brutal and progressive telomere DNA shortening. Telomere replication at the leading or lagging strands is not similarly regulated. At the G′-rich lagging strand, due to the end-replication problem, the parental strand is not completely replicated, triggering a 3′-overhang. At the leading strand, there is a blunt end or 5′protruding that is eroded in a manner coupled to replication by the TRF2-dependent recruitment of the 5′-exonuclease Apollo. Next, at both strands, the Exo1 exonuclease prolongs the 3′-overhang. Finally, POT1 recruits the CST complex (CTC1-STN1-TEN1) to replenish the 5′-strands. The brutal appearance of critically short telomeres that cannot be explained by the progressive telomere shortening was first observed in yeast [11]. They are likely to stem from intra-telomeric recombination [12,13] or the collapse of DNA replication forks. Indeed, it is known that telomeric repeats can form recombinogenic t-loops [14,15] and are difficult to replicate [16,17,18].

**Figure 2 cells-10-01156-f002:**
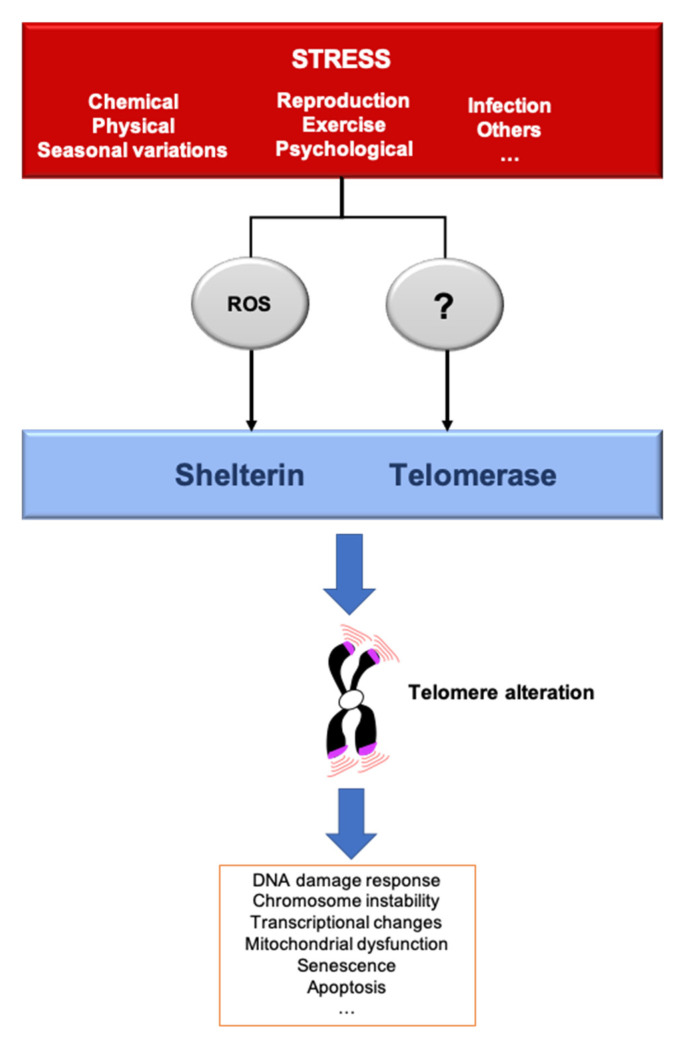
Stressors affect telomere homeostasis. Several stresses such as changes in pH or temperature, exercise, irradiation, pathogens infection, reproduction/breeding efforts, exposure to chemicals or psychological pressure, among others, can affect telomere integrity. The most described mechanisms induced by several of these stressors include telomerase inhibition and an increase of ROS, that preferentially oxidizes the telomeric DNA, a process that we dubbed teloxidation [43]. Upon challenge, telomeric alterations induce the DNA damage response activation (DDR), senescence, transcriptional changes and mitochondrial dysfunction.

**Figure 3 cells-10-01156-f003:**
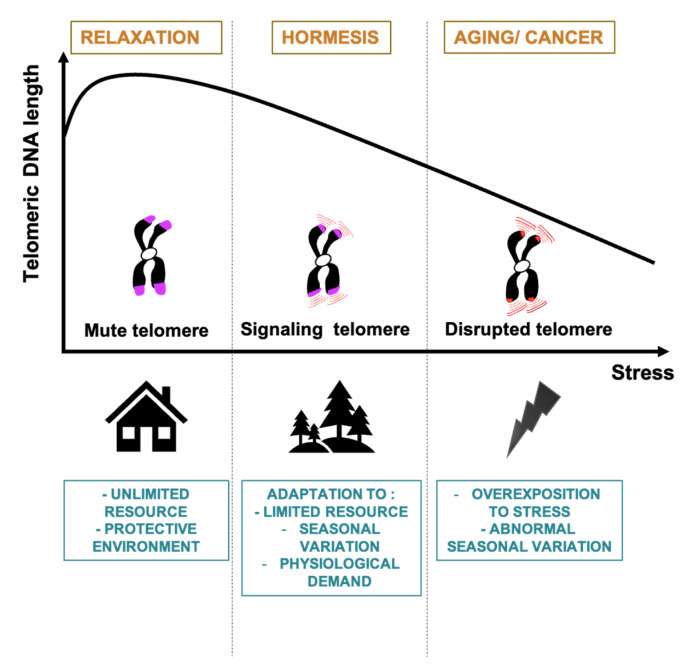
The telo-hormesis hypothesis. Under exposure to predicted stresses (such as regular seasonal fluctuations and limited resource) or normal/physiological demands, telomere integrity might be affected in order to trigger a hormetic signal through transcriptional changes and mito-hormesis. To stop an early oncogenic event, this hormetic signaling can also result in cellular senescence. Under unexpected, overexposure and acute stresses, telomere integrity can be dramatically disrupted leading to detrimental effects with an excessive chromosome instability resulting in premature aging and cancer development. In contrast, in protective, unchallenged conditions, one can speculate that telomeres are in a “mute” state that just protect chromosome ends but without leading to any beneficial or detrimental signaling, resulting in the “relaxation” of the TL regulation processes, possibly leading to heterogenous and long telomeres, such as those present in laboratory mice.

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
