# Peer review of "The Power of Stress: The Telo-Hormesis Hypothesis"

_cells, 2021, doi:10.3390/cells10051156_

Round 1
Reviewer 1 Report
« The power of stress: the telo-hormesis hypothesis » by Maria Sol Jacome Burbano and Eric Gilson.
This review fits very well with this special issue in honor to Miroslav Radman. Among his tremedous contributions in Science, Miroslav has been a pioneer in the concept and characterization of adaptative stress response (SOS response theory during his PhD! and later on adaptative mechanisms of highly resistant bacteria to radiation). This review discussing on the impact of stress on telomeres and adaptative role of telomeres shortening fits also perfectly with Miroslav’s interest in aging.
In summary in this review, the authors introduce the “telomere biology”, how different stress can alter telomeres, and discuss the hypothesis of a stress-induced telomere signaling in hormesis.
The authors advocate that “telo-hormesis” is an enhanced strategy to combat the stress response, and this based on a rich study of litterature…
They postulate that short telomeres may improve survival and fitness during exposure to stress.
This review is very interesting, especially chapter discussing how telomeres length are affected upon different stress. Importantly, this review invites us to revisit the view that telomere shortening was selected as an anti-cancer response. This concept is of importance to propose and to guarantee new therapeutical agents to fight aging.
This review must be published but could be improved and revised a little bit to be definitively accepted for publication.
1- A lot of observations are described but underlying molecular mechanisms are often not sufficiently described. This is important for a better understanding and also to provoke to the reader a fascinating scientific thinking, in order to consider novel hypotheses to progress in this field, and also to reach interest of scientists outside the telomere field.
For instance:
- The description of brutal shortening by fork collapse or intra-telomeric recombination should be described in text or in Figure 1 legend.
- How rapid shortening could be a rapid response of stress (or an advantage for tumor cell growth?) What could be the underlying mechanisms?
- Description of the molecular mechanism regulating long-range chromatin-loops and depending of TL and shelterin subunits, playing a crucial role in gene regulation could be better described.
2- Iconography and figures legends should be improved to fit more with text.
Figure 1: This figure could be improved, in legend or in the figure body, name of proteins in shelterin complex should be noted. Drawing should correspond or be a better illustration of the sentences in text, as for : “TRF1 manages classical stalled forks to ATR activation and TRF2 manages topologically-constrained stalled forks » [21,22] (Figure 1).
3- The impact of nutrition or physical exercise (sport) on telomeres length should be also mentioned.
References could be useful:
DOI: 10.3945/jn.116.230490 on impact of caffeine
doi: 10.1016/j.bbi.2012.09.004 and more recently doi: 10.1186/s12887-020-02487-x on impact of omega 3 fatty acid
Physical activity a may be beneficial for telomere length maintenance PMID: 28410238 and PMID:30496493 impact of aerobic training on telomerase activity.
Thus, the notion of reversibility irreversibility of telomeres shortening may be discussed.
4- Line 165 to 183: In addition to the impact of inflammation on TL, the link between telomere maintenance defects and increased risk of chronic inflammatory conditions in humans should be discussed
References should be included, such as YAP1 activation upon telomere dysfunction to drive tissue inflammation (doi: 10.1038/s41467-020-18420-w) or link with autophagy (doi: 10.1038/s41586-019-0885-0).
5-It could be interesting to discuss on the “Wonbat riddle”: one of the only species using the ALT lengthening pathway. What could be the evolutionary advantage? Since ALT could instead promote cancer…
Minor points:
Line 32-43: Missing some references
Line 72: Typo “protruding”
Line 76: Please add references
Line 257: other example of short-living organism? What about mouse in nature?
Line 350: “Accordingly, protection against melanoma development might be a driver of short TL in European populations compared to Sub-Saharan Africans [92].” Could we imagine than UV damage may affect in more extensive way European DNA than Sub-Saharan Africans DNA????
Author Response
We thank reviewer #1 for his/ her positive appraisal and deep reading of this review.
Please see the attachment.

Reviewer 2 Report
The manuscript by Burbano and Gilson discussed the hypothesis of a stress-induced telomere signaling involved in hormetic effects, they termed as telo-hormesis. Hormesis is defined as an adaptive response to low stress exposure that confers further protection against high stress exposure. Authors’ hypothesis is thus that telomeres participate to such adaptive responses.
After recalling the basic molecular mechanisms of telomere biology, they show how different stress —ie. inflammation, psychological, chemical and physical stress— can alter telomere structures and functions, mainly leading to telomere shortening. They described then the telomere shortening associated to reproduction. Finally, a relatively short part is dedicated to the telo-hormesis hypothesis, discussing successively the roles of telomeres in tumor prevention, in transcriptional regulation (telomere position effects) and in mito-hormesis, one of the most detailed pathway of hormesis which is associated to mitochondrial stress.
While the hypothesis of telo-hormesis is seducing, its description lacks clarity being sometimes confusing. This needs to be strengthened before publication.
Main Concerns
- Authors’ hypotheses concerning possible telomere hormetic effect in cancer prevention are not really clear. Moreover, telomere shortening is presented as a possible hormetic effect in conditions of limited resources, wild environment etc… leading to a functional advantage by preventing tumor formation. However, the authors fail to show that this is really a hormetic effect (ie conferring further protection against higher stress exposure), rather than a “classic” stress response, that could be beneficial or detrimental depending on the context.
- Quite surprisingly, the hypothesis of hormesis leading to repressive effects on telomere shortening as reported by Seiichi Yokoo ll Biochem. 2004 is not discussed.
- The descriptions of the potential links between hormesis and the telomere position effect and those between telomeres and mito-hormesis are very interesting, but very short. They should be better structured and strengthened.
Minor concerns
- Lane 257: The reference to laboratory mouse that have long telomeres is irrelevant here, since it has been shown that there is no correlation of telomere length with lifespan in a number of mouse strains and wild type mice having short telomeres (Hemann and Greider, 2000).
- Line 416: How olfactory receptor genes may be involved in an hormetic effect?
- The discussion of the telomere cost due to reproduction is sometimes confusing. A separate chapter dedicated to lifespan and telomere lengths would have been more appropriate.
- Line 323: The definition of telo-hormesis as “an enhanced strategy to combat the stress response” seems not really appropriate.
- Figure 1: No progressive telomere shortening is shown.
- Figure 3: This figure is confusing and fails to give a clear overview of authors’ hypothesis. For instance, what exactly means telomere integrity here, is it telomere length? Why (and how!) telomere integrity may increase in conditions with unlimited resource and protective environment?
- Line 468: Ref 8 should be edited.
Author Response
We thank reviewer #2 for the time expend on our work and for his / her valuable comments. We agree our review might be confusing and we try to better present our ideas in the revised manuscript.
Please see the attachment.

Reviewer 3 Report
This is a very nice, well written review that proposes hormetic effects of telomere length modulation on health and viability within a cell or organisms. Although speculative, the authors’ telo-hormesis hypothesis is intriguing and could indeed posit the existence of programmed telomere change requirement during development. However, the authors have to pay more attention when they define hormesis (a dose-response phenomenon characterized by low-dose stimulation and high-dose toxicity) particularly when compared to adaptive response (in which exposure to minimal stress could result in increased resistance to higher levels of the same or to other types of stress later). These two terms are erroneously used as synonyms through the text (especially on pag 9, line 330)
Minor concerns
On page 2, line 66. It should be mention that telomere length in Drosophila, in addition to retrotransposistion, can rely also on gene conversion and recombination events (McGurk et al, 2021; Genetics 217; Kurn and Begun, 2008; Genetics 179; Cacchione et al., 2020; JMB 432).
On page 10, Line 398. Refs 116-119 do not refer to TPE in Drosophila or Trypanosoma, as indicated in the corresponding sentence.
In the section “4: Telomere cost due to reproduction” I would suggest the authors to include the evidence of a correlation between long telomeres and reduced fertility on Drosophila melanogaster (Walter et al., 2007: Chromosoma 116)
Author Response
We thank reviewer #3 for his / her valuable remarks.
Please, see the attachment.

Round 2
Reviewer 2 Report
The authors answered my concerns quite satisfactorily.